# LIFT: Enhancing Long-Context Reasoning in Large Language Models via Long Input Fine-Tuning

## Abstract

Long context understanding remains challenging for large language models due to their limited context windows. This paper introduces **Long Input Fine-Tuning (LIFT)**, a novel framework for long-context modeling that can enhances the long-context performance of arbitrary short-context LLMs by dynamically adapting their parameters to the given long input. Importantly, rather than endlessly extending the context window size to accommodate increasingly longer inputs **in context**, LIFT stores and absorbs the long input **in parameters**. To achieve this, LIFT employs a novel methodology that leverages synthetic tasks generated by LLMs to train models. By fine-tuning the long input into model parameters, LIFT allows short-context LLMs to answer questions even when the required information is not provided in the context during inference. In contrast to conventional approaches that rely on continued pretraining on long contexts, LIFT enables the models to achieve true comprehension of long contexts, moving beyond mere memorization. We further provide a comprehensive analysis of LIFT's strengths and limitations in long-context understanding, discuss its feasibility for large-scale real-world deployment, and highlight valuable directions for future research.

## 1 Introduction

Recent advances in large language models (LLMs), such as OpenAI o1 (OpenAI, 2024) and DeepSeek-R1 (DeepSeek-AI et al., 2025), have reshaped natural language processing, enabling state-of-the-art performance across tasks like text generation, translation, summarization, while substantially improving performance on challenging reasoning tasks. However, despite these advances, long-context reasoning remains a fundamental challenge for LLMs. Long sequences, which can span up to millions of tokens, are common in real-world applications, including long books (Kočiský et al., 2018), accounting documents (Li et al., 2024), high-resolution videos (Wu et al., 2024; Tapaswi et al., 2016), and audio signals (Yang et al., 2024).

Limited by the positional embeddings, LLMs often struggle to generalize beyond the sequence lengths seen during training, resulting in an upper bound on the input length they can process, a.k.a. the context window size. Extending context windows allows LLMs to capture dependencies across larger text spans and improves coherence, understanding, and accuracy in tasks that require reasoning over extended inputs. However, as context lengths increase, the computational complexity of the self-attention mechanism (Vaswani, 2017) grows quadratically, which limits models' ability to process long inputs. Additionally, storing a large number of attention weights and intermediate states like KV cache places a heavy burden on hardware resources. Moreover, it is challenging to capture long dependencies among pieces of information scattered throughout long inputs while performing further comprehension and reasoning. Due to the limitation of context windows, LLMs can hardly capture the overall information about a user's query history or task input, resulting in suboptimal performance.

To address these challenges, researchers have developed various techniques to improve the long-context abilities of LLMs. Retrieval-Augmented Generation (RAG) (Lewis et al., 2020; Xu et al., 2023) and prompt compression (Jiang et al., 2023) aim to preprocess long inputs within a limited short context window by adaptive retrieval or text compression (El-Kassas et al., 2021). However,

Figure 1: An overview of the LIFT workflow. The process begins by splitting a long input (e.g., a document) into sentences, which are then sent to a local/remote LLM server to generate synthetic tasks in parallel. These tasks are used to fine-tune a short-context LLM, yielding a LIFTed LLM that can answer questions without directly accessing the original input.

Table 1: Comparison of conventional long context understanding approaches with LIFT.

|  | RAG | Long-context post-training | LIFT |
|---|---|---|---|
| Knowledge storage | External database | Context window | Parameters |
| Input length | Infinite | Limited | Infinite |
| External-retriever-free | ✗ | ✓ | ✓ |
| Long-context-processing-free | ✓ | ✗ | ✓ |

the effectiveness of these methods depends on the precision and relevance of the contextual information provided within the context window. It will lead to further hallucinations when noisy, ambiguous, or conflicting information is provided. Long-context post-training focuses on fine-tuning pretrained LLMs on corpora of long texts to extend their context windows (Chen et al., 2023; Peng et al., 2023) and is more frequently used in more recent works. However, the adaptation process comes with significant costs in terms of both training data and computational resources. Additionally, with the extended context window, the cost of processing and generating long texts grows quadratically with the input length. Finally, despite the extension, the context windows of these LLMs remain finite, preventing them from generalizing to inputs of unbounded length.

To overcome these limitations and enable efficient reasoning over long inputs, in this paper, we present **L**ong **I**nput **F**ine-**T**uning (LIFT), a novel framework designed to enhance the long-context capabilities of arbitrary (short-context) models by directly adapting model parameters to the long input. Table 1 provides a comparison of LIFT with major existing approaches, including long-context post-training and retrieval-augmented generation (RAG). Our approach has the following advantages:

- **Efficient fine-tuning and decoding.** LIFT dynamically adapts an LLM to newly introduced long inputs as fresh knowledge by adjusting its parameters. To enhance comprehension of the long input, we generate synthetic QAs according to its sentences, and supervised fine-tune (SFT) the LLM on batches of QAs using data parallelization to ensure efficient adaptation. Compared to a long-context model, LIFT avoids the quadratic complexity w.r.t. the context length, maintaining the fast decoding speed of a short-context LLM. Experiments confirm its great efficiency advantages.

- **Great improvement on popular long-context tasks.** Our evaluations on well-acknowledged long context benchmarks show that LIFT consistently benefits diverse downstream tasks like long/short-dependency question answering (QA) and summarization. For example, on the challenging long-dependency QA tasks of LooGLE (Li et al., 2023), the "LIFTed" Llama-3-8B-Instruct model achieves an accuracy of 29.97%, significantly outperforming its pure ICL counterpart without LIFT which achieves only 15.44% accuracy.

## 2 RELATED WORK

**Long-context post-training and efficient architectures.** Existing LLMs mostly rely on pure in-context learning (ICL) for long-context understanding. However, it is challenging for short-context models to process inputs longer than their context window sizes due to unseen positional embeddings during pretraining, resulting in extremely poor performance on downstream tasks. Therefore, a common practice is to further post-train LLMs on a huge corpus of long texts. Despite the effectiveness, long-context post-training often requires tremendous computational cost. To cope with the problems, many works have been developed to accelerate the process of long-context training with efficient Transformer. Sparse attention (Kitaev et al., 2020; Wang et al., 2020; Beltagy et al., 2020) reduces memory and computation costs by using local windows or strided attention, allowing to focus on the most relevant inputs for given tasks. Linear attention (Shen et al., 2021) reduces the quadratic computation to linear by approximating self-attention with kernel functions or low-rank representations. Other alternatives for Transformer like state-space models (SSMs) (Gu & Dao, 2023) are recently proposed for efficient training based on dual representations. However, such efficient architectures have not been widely adopted, largely due to their performance degradation compared to standard Transformers and their incompatibility with existing hardware accelerators. As a consequence, in this work, we focus on the conventional self-attention architecture (Vaswani, 2017) which is most widely used in current LLMs to validate the effectiveness of LIFT.

**Retrieval-Augmented Generation (RAG).** RAG (Lewis et al., 2020) improves the performance of long-context understanding by integrating LLMs with external data sources for retrieval (Xu et al., 2023; Jiang et al., 2024b; Wang et al., 2024a; Jin et al., 2024), thereby avoiding the need to feed the entire long input. Its performance heavily relies on the quality of retrieved content, which must be relevant and concise enough to fit within models' short context windows. RAG can experience significant performance degradation or hallucination issues when the retrieved context is inaccurate or mismatched. A comparison of our LIFT with RAG and long-context adaptation is in Table 1.

**Memory-augmented LLMs.** A line of work (Wang et al., 2023; 2024c; 2025) explore augmenting LLMs with a memory module. Compared to RAG, which builds an offline database and retrieves from it during inference, memory-augmented LLMs emphasize continual updates of the memory module, enabling them to process long inputs sequentially. Wang et al. (2023) design a memory module that memorizes the hidden states as the LLM processes a long input and exponentially forgets past knowledge. While most memory-augmented LLMs memorize hidden states with an external module, our work explores directly storing incoming knowledge within model parameters.

**Test-time training.** Test-time training (TTT) (Liu et al., 2021; Gandelsman et al., 2022; Osowiechi et al., 2023; Hong et al., 2023; Wang et al., 2024b) has emerged as a promising approach to adapt models to unseen data distributions during deployment, leveraging test data to fine-tune the model at inference time. Recent works have applied similar ideas to improve model adaptability when dealing with lengthy, context-rich inputs (Sun et al., 2024; Behrouz et al., 2024), yet focus on proposing new architectures to replace Transformer and require pretraining from scratch. Our work, in contrast, focuses on improving arbitrary pretrained models' long-context capabilities by fine-tuning them on the long input, which is not restricted to specific models or layers. Wang et al. (2024b) propose TempLoRA, which is closely related to LIFT. It explores how TTT can enhance LLMs in long generation tasks such as novel writing and translation through iteratively fine-tuning a LoRA adapter to memorize the previously generated tokens. While sharing a similar idea to store context knowledge in LLM parameters, LIFT focuses on long-context reasoning tasks like long-context question-answering, which present greater challenges for comprehension. Our experiments indicate that continued pretraining alone is inadequate for achieving true comprehension; by contrast, LIFT leverages synthetic tasks, which have been shown to be effective in enabling LLMs to better understand long inputs.

## 3 MOTIVATING EXPERIMENT

A naive way to memorize the long input into parameters is to continue pretraining the model on the long input using next token prediction objective. However, in this section, we present a motivating experiment suggesting that continued pretraining is insufficient for achieving true comprehension.

Table 2: Results of the motivating experiment.

| Epoch | Loss | Question-answering | Continued writing |
|-------|------|--------------------|--------------------|
| 0 | 2.100 | San Francisco! There are so many amazing things to do and see in this vibrant city. Here are some of... | to explore the city's many neighborhoods, each with its own unique character and charm.... |
| 2 | 1.474 | San Francisco! There are so many amazing things to do in this vibrant city. Here are some of the best... | to take a walk across the Golden Gate Bridge. The views of the city, the bay, and the Pacific Ocean... |
| 4 | 0.758 | San Francisco is a vibrant and diverse city with a wide range of activities and attractions to... | eat a sandwich and sit in Dolores Park on a sunny day. ... |
| 6 | 0.475 | San Francisco is a vibrant and diverse city with a wide range of activities and attractions to... | eat a sandwich and sit in Dolores Park on a sunny day. ... |

We know that pretrained LLMs encode the knowledge from their pretraining corpora into their parameters and can reason over the internalized knowledge. Thus, one might expect that continued pretraining on a specific input would enable them to also understand the input. Here we define continued pretraining as fine-tuning LLMs on the following language modeling objective:

$$\mathcal{L}_{\text{CP}}(\mathbf{x}; \theta) = -\sum_{i=1}^{|\mathbf{x}|} \log f_\theta(\mathbf{x}_i | \mathbf{x}_{<i}),$$

where $\mathbf{x}$ is the input text and $f_\theta$ represents an LLM that predicts the distribution of the token $\mathbf{x}_i$ given the previous tokens $\mathbf{x}_{<i}$.

We conduct a preliminary experiment on a simple task—given the sentence "The best thing to do in San Francisco is eat a sandwich and sit in Dolores Park on a sunny day." as context, the model is asked a simple question, "What is the best thing to do in San Francisco?" (Kamradt, 2023). We evaluate Llama-3-8B-Instruct on this task. When the sentence is provided within the context window, the model can easily generate the correct answer, "Eat a sandwich and sit in Dolores Park on a sunny day.", with its in-context learning ability.

Then, we conduct continued pretraining on this sentence, where we fine-tune Llama-3-8B-Instruct on the sentence for varying numbers of epochs and inspect the final training losses. With the updated models, we ask the same question "What is the best thing to do in San Francisco?" and present the model answers in Table 2. The results are surprising, as no matter how low the loss is, the model **fails to answer faithfully according to the sentence it just trains on**. As a sanity check, we also prompt each model with "The best thing to do in San Francisco is to" and check the continued writing results. As we can see, models with sufficient training epochs easily recite the correct sentence.

This interesting phenomenon suggests that memorization does not mean comprehension, and naively training on the long input does not guarantee the model can truly understand the knowledge and effectively leverage it in reasoning. It also suggests that fine-tuning has completely different dynamics from pretraining—we cannot expect models to leverage its fine-tuned knowledge similarly to its pretrained knowledge. This experiment motivates us to design different training objectives as introduced in the next section.

## 4 METHOD

In this section, we present LIFT, a framework designed to enhance the long-context understanding of LLMs through long input fine-tuning (Figure 1). We begin by describing our method for generating synthetic tasks, followed by a discussion of our pipeline design aimed at accelerating the process for practical, real-world deployment.

Table 3: Demonstration of the synthetic tasks on LooGLE.

| Sentence | Synthetic tasks |
|----------|-----------------|
| Picardo was born in Jerez de la Frontera, in the Province of Cádiz in Andalucía, Spain on 18 June 1919. | **Q**: Can you name the specific city where Picardo was born? 
 **A**: Jerez de la Frontera |
| The first workers to arrive lived in 125 US Army pyramidal tents with wooden floors and sides while they erected the first barracks. | **Q**: How many US Army pyramidal tents were set up for the first workers to live in? 
 **A**: 125 |
| The match was drawn 10–10. | **Q**: Did the match end in a tie? 
 **A**: Yes |

## 4.1 SYNTHETIC TASK GENERATION

As illustrated in the motivating experiment, successfully memorizing the lengthy input does not necessarily indicate that the model can reason effectively based on it.

Our approach is motivated by two observations. First, Jiang et al. (2024a) argue that knowledge is expressed more directly through question-answer pairs, whereas raw documents are often harder for models to understand. Second, educational research has long demonstrated that reading while questioning is an effective strategy for enhancing human comprehension of knowledge (Robinson, 1946). In summary, representing knowledge in the form of question–answer tasks can make it more accessible to both LLMs and human. Drawing inspiration from both findings, we propose to leverage LLMs to generate synthetic tasks in the form of **question–answer pairs**, thereby transforming long inputs into a representation that facilitates internalization and reasoning.

Formally, given the long input $\mathbf{x}$, we prompt an LLM (hereafter referred to as the generator, to distinguish it from the target LLM trained by LIFT) to generate question-answer pairs, $(\mathbf{q}_i, \mathbf{a}_i)_{i=1}^{m}$, based on $\mathbf{x}$. These QAs can be simple details such as specific people, time, locations of events, or more general reading comprehension ones. In practice, to avoid processing long sequences, we split $\mathbf{x}$ into sentences and prompt the generator to synthesize question-answer pairs for each sentence. Representative demonstrations are provided in Table 5 where we adopt Qwen2.5-72B-Instruct as the generator and the synthetics tasks are generated given the corresponding sentences.

We train the target LLM on the synthetic tasks through supervised fine-tuning, using the following objective:

$$\mathcal{L}_{\text{syn}}((\mathbf{q}_i, \mathbf{a}_i)_{i=1}^{m}; \theta) = -\sum_{i=1}^{m} \log f_\theta(\mathbf{a}_i \mid \mathbf{q}_i). \tag{1}$$

There are no strict restrictions on how $(\mathbf{q}i, \mathbf{a}i)_{i=1}^{m}$ are synthesized from $\mathbf{x}$. In practice, one may design tailored prompts or pipelines to generate synthetic tasks aligned with specific downstream applications. For instance, in novel comprehension, the generator can be prompted to focus on aspects such as the timeline, main characters, and other salient elements. In our case, however, since we aim at general long-context tasks, we deliberately avoid introducing inductive biases into synthetic task generation. Furthermore, because a single sentence extracted from a long input may contain incomplete information (e.g., pronouns with unresolved references), we provide each sentence to the generator along with a short preceding paragraph as context. This ensures that the generator can fully interpret the sentence, extract the relevant information, and represent it in the form of question–answer pairs. For the detailed prompts we used, please refer to Appendix A.

## 4.2 DESIGN FOR EFFICIENT DEPLOYMENT

The LIFT pipeline consists of two major components: synthetic task generation and fine-tuning. To enhance its efficiency, especially to reduce the first-token latency, we introduce several designs that jointly accelerate both synthetic task generation and fine-tuning.

First, given a fixed token budget per sentence, we generate multiple short question–answer (QA) pairs for each sentence instead of a single long QA pair. From a computational perspective, training

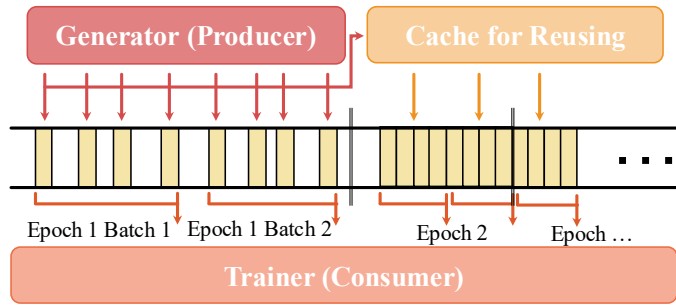

Figure 2: A demonstration of the workflow of the asynchronous producer–consumer pipeline.

on several short QA pairs is less complex and more efficient than training on one long QA pair, while still capturing the essential information conveyed by the original sentence. Specifically, suppose we generate $m$ QA pairs, each of length $l$. Training on these QA pairs yields a complexity of $\mathcal{O}(ml^2)$. In contrast, suppose we generate a single QA pair of length $ml$. Training on it yields a complexity of $\mathcal{O}(m^2l^2)$. Thus, dividing a long QA into multiple shorter ones substantially reduces the training overhead. Moreover, pretrained LLMs like Qwen2.5-72B-Instruct are capable of generating multiple QA pairs that each highlight different aspects of the given sentence, thereby preserving the richness and diversity of the synthetic tasks.

Finally, we design an asynchronous producer-consumer pipeline that jointly accelerates the generation-training workflow. In this pipeline, the generator acts as a producer that continuously outputs QA pairs, while the trainer acts as a consumer that retrieves the generated QA pairs to construct mini-batches for fine-tuning. The workflow of this asynchronous producer–consumer pipeline is illustrated in Figure 2. If insufficient tasks are available to fill a batch, the trainer blocks until new tasks arrive, whereas the generator operates independently of the trainer's progress. With the parallelized generation optimization mentioned above, we allow the production rate matches the consumption rate, thereby reducing idle time in the pipeline. Importantly, the trainer only experiences blocking in the first epoch. Once all synthetic tasks are generated and cached, subsequent epochs can directly fetch data from the cache without delay.

Together, these designs substantially accelerate the LIFT pipeline, reducing the overhead of synthetic task generation and fine-tuning to seconds for moderately long input and thus greatly reducing the first-token latency.

## 5 EXPERIMENTS

### 5.1 SETUP

**Dataset and metrics.** We evaluate LIFT on two complementary long-context benchmarks that together provide a comprehensive assessment of the capabilities of LIFT. We use NIAH (Kamradt, 2023) as a basic test of long-text processing, focusing on whether models can effectively retain and recall information from extended inputs. In contrast, we adopt LooGLE (Li et al., 2023) as a more advanced benchmark that challenges models with complex long-text reasoning tasks. This combination allows us to examine both the basic and higher-level aspects of long-context understanding.

Specifically, the NIAH benchmark characterizes each test case with two attributes: the document length $L$ and the insertion depth $D$ (expressed as a percentage). Specifically, it inserts the needle sentence "The best thing to do in San Francisco is eat a sandwich and sit in Dolores Park on a sunny day." into a document of length $L$ at position $D$, and uses this document as the context. The model is then required to answer the question "What is the best thing to do in San Francisco?" based on the provided context. As $L$ increases, the test becomes more challenging, while varying $D$ evaluates whether the model suffers from the lost-in-the-middle problem (Liu et al., 2023a).

LooGLE benchmark consists of two subtasks, ShortQA and LongQA. In both subtasks, a test case provides the model with a long document and requires the model to answer several questions based

Table 4: Performance of different methods on LooGLE based on the LLama-3-8B-Instruct model. We evaluate the accuracy of the methods on LooGLE short-dependency QA (ShortQA) and long-dependency QA (LongQA). Comprehension & reasoning, multiple info retrieval, computation, and timeline reorder are the subtasks in LongQA and we evaluate the accuracy on each of them.

| Methods | ShortQA | LongQA | Comprehension & Reasoning | Multiple info retrieval | Computation | Timeline reorder |
|---|---|---|---|---|---|---|
| MemoryLLM | 33.06 | 20.44 | 29.31 | 15.53 | 8.00 | 18.14 |
| LlamaIndex | 41.93 | 21.07 | **33.00** | 17.11 | 12.00 | 9.77 |
| ICL | 44.49 | 15.44 | 25.37 | 15.26 | 5.00 | 1.86 |
| LIFT with 5QA | 45.67 | 26.79 | 29.80 | 21.58 | 14.00 | 36.28 |
| + segmented LM | 44.08 | 26.61 | 27.83 | 20.79 | 14.00 | 40.47 |
| + truncated ICL | 49.31 | 27.52 | 31.28 | 22.37 | 10.00 | 37.67 |
| + both | 50.28 | 26.70 | 29.31 | 23.42 | 11.00 | 34.88 |
| LIFT with 10QA | 52.69 | 27.25 | 27.83 | 22.63 | **16.00** | 39.53 |
| + segmented LM | 54.07 | 26.70 | 29.56 | 22.37 | 15.00 | 34.42 |
| + truncated ICL | **56.43** | **28.52** | 28.82 | **23.68** | 14.00 | **43.26** |
| + both | 55.10 | 24.34 | 26.11 | 20.53 | 11.00 | 33.95 |

on the document. The tasks in ShortQA require the model to extract information from specific sentences, while the tasks in LongQA require the model to gather information across the entire document. In general, LongQA is harder than ShortQA.

The evaluation metrics are consistent with those used in the original benchmarks. For NIAH, a response is considered as correct if the keywords, "sandwich", "Dolores Park", and "sunny", appear in the response. For LooGLE, since most automatic evaluation metrics are sensitive to semantic expression, output format, and length, we utilize GPT-4.1-nano (OpenAI) as recommended in the paper to judge whether the two answers are semantically the same or not, noted as GPT4-score. It has been proven to exhibit high consistency with human evaluation and can serve as a reliable annotator to a great extent (Suri et al., 2023; Liu et al., 2023b; Zheng et al., 2023). The prompts implemented can be found in Appendix B.

**Baselines.** We select `Llama-3-8B-Instruct` with 8k context windows as the base LLMs, and compare the performance with and without LIFT. We also compare LIFT with a representative RAG framework LlamaIndex (Liu, 2022), as well as MemoryLLM (Wang et al., 2024c), a memory-augmented LLM.

**Settings.** We use **ICL** to denote truncating the long input by retaining only the beginning and end of texts within the context window of the base model. We use **LIFT** to denote fine-tuning the base LLM using the LoRA (Hu et al., 2022) adapter on the synthetic tasks. Notice that we prompt the generator to synthesize $m$ question-answer pairs for each sentence and we can adjust $m$ to tradeoff efficiency and performance. For **MemoryLLM**, we reproduce its implementation and evaluate it on LooGLE. For **LlamaIndex**, we adopt `bge-small-en-v1.5` (Xiao et al., 2023) as the embedding model. **Truncated ICL** refers to the practice of appending a portion of the original input—within the constraints of the context window length—to the question in order to improve the language model's comprehension. **Segmented LM** indicates that segmented sections of the original input are incorporated into the training data.

### 5.2 RESULTS ON NIAH

As shown in Figure 4, LIFT achieves perfect accuracy on the NIAH benchmark. In comparison, in-context learning (ICL) attains nearly perfect accuracy but fails in one case to extract the required information. These results demonstrate that LIFT can reliably retain and retrieve information from long inputs. Note that LIFT only generates 1 QA per sentence to reach full accuracy. As an illustrative example, Table 5 presents synthetic QAs generated by LIFT using Qwen2.5-72B-Instruct generator for the NIAH needle sentence. We observe that the generator accurately captures all three key phrases. Moreover, LIFT does not simply memorize the generated synthetic tasks, as the synthetic questions differ literally from the actual evaluation question. This indicates that LIFT leverages the

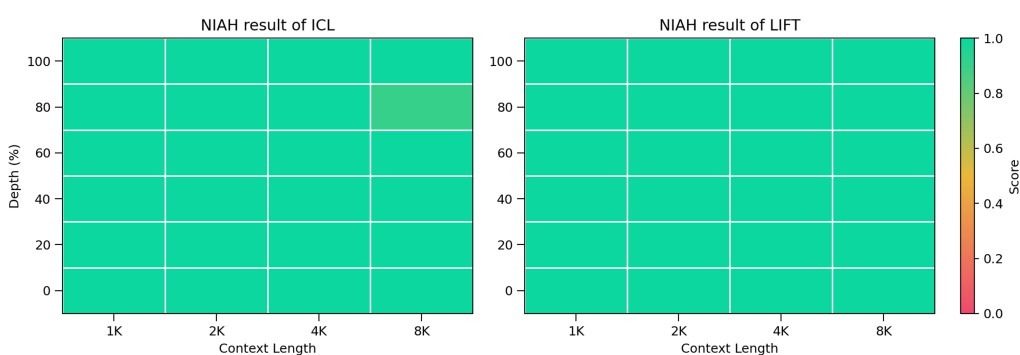

Figure 3: The NIAH result of LIFT.

Table 5: Demonstrations of the synthetic tasks generated for the NIAH needle.

| Question | Answer |
|---|---|
| What is recommended as the best activity to do in San Francisco according to the sentence? | The best thing to do in San Francisco is eat a sandwich and sit in Dolores Park on a sunny day. |
| What is the best thing to do in San Francisco according to the sentence? | The best thing to do in San Francisco is eat a sandwich and sit in Dolores Park on a sunny day. |
| What is recommended as the best thing to do in San Francisco according to the sentence? | The best thing to do in San Francisco is eat a sandwich and sit in Dolores Park on a sunny day. |

Table 6: The first-token latency of LIFT with varying input lengths.

| Input length | 1000 | 2000 | 4000 | 8000 |
|---|---|---|---|---|
| First-token latency (sec) | 1.98 | 3.43 | 5.90 | 9.92 |

synthetic QA tasks to genuinely internalize new knowledge and apply it for reasoning, rather than relying on rote memorization.

We perform additional efficiency experiments to verify the efficiency of LIFT. By transferring input tokens into LLM parameters, LIFT **avoids the need to compute the attention score over all tokens in the long input** when generating a token. Consequently, the decoding speed of LIFT is expected to be much faster than that of long-context ICL which requires attention computation over the entire long context. We measure the total time cost (including the startup fine-tuning time of LIFT) of generating $x$ tokens with 16k tokens as the input. Empirically, as illustrated in Figure 4, LIFT starts to outperform ICL in decoding time when generating more than 2k tokens. This is because, LIFT only needs to fine-tune on the long input once, and later *fully becomes a short-context model* with very short decoding time per token. In contrast, ICL puts all the long input in the context, and every new token generation needs to compute the attention of the last token to all the previous tokens, incurring great latency in every new token generation. We further present the first-token latency of LIFT with varying input lengths in Table 6. As we can see, LIFT takes less than 10 seconds to absorb 8k input and start to output the first token. For moderately long input (such as 1k tokens), the latency even decreases to 2 seconds, demonstrating that LIFT can be deployed in practical scenarios.

### 5.3 RESULTS ON LOOGLE

**Overall performance.** As shown in Table 4, LIFT consistently outperforms all baseline methods on the LooGLE benchmark. On the ShortQA subtask, LIFT with 10 generated QA pairs per sentence

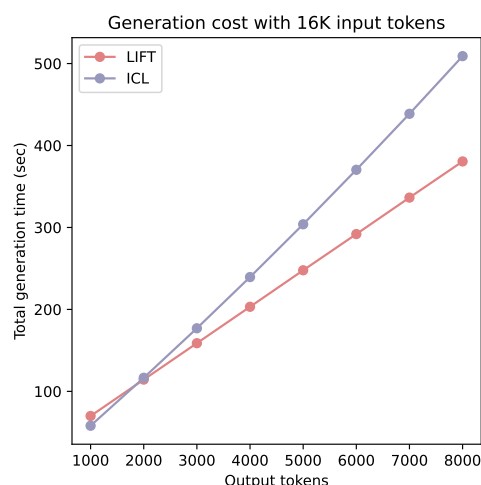

Figure 4: Total generation time with varying output lengths given 16K input tokens.

achieves accuracy above 50%, exceeding all baselines by a substantial margin. On the more challenging LongQA subtask, LIFT surpasses the strongest baseline, MemoryLLM, by 8%, representing a significant improvement in long-context reasoning performance. Overall, these results demonstrate the effectiveness of LIFT in both information extracting capability and reasoning capability.

Specifically, we observe that increasing the number of synthetic tasks improves performance on ShortQA, but raising it from 5 to 10 per sentence yields no further gains on LongQA. This difference can be explained by the difference of the two subtasks. ShortQA primarily evaluates information extraction, and generating more synthetic tasks increases coverage of sentence-level details, thereby boosting performance. In contrast, LongQA requires the model to integrate information across the entire document and perform reasoning. Since additional synthetic tasks mainly enhance local information coverage rather than information association ability, they provide little benefit for LongQA.

**Performance across the four categories of LongQA.** To provide a more fine-grained evaluation, the LongQA subtask is further divided into four categories, comprehension & reasoning, multiple-info retrieval, computation, and timeline reorder, each designed to assess a distinct aspect of long-context capability. LIFT yields the largest improvements on the multi-information retrieval and timeline reorder categories, as the synthetic tasks primarily assist the model in better understanding and retaining the information provided in the document. In contrast, solving tasks in computation and comprehension & reasoning requires the model to combine the external knowledge distilled from synthetic tasks with its own internal reasoning capabilities in order to arrive at correct answers.

## 6 CONCLUSION

In this paper, we proposed **L**ong-**I**nput **F**ine-**T**uning (**LIFT**), a novel framework designed to enhance the long-context understanding of LLMs. LIFT dynamically adapts model parameters to long inputs and leverages the resulting in-parameter knowledge to improve performance on long-context tasks. Our experiments show that LIFT achieves perfect accuracy on the NIAH benchmark and yields significant improvements on the more challenging LooGLE benchmark.

Beyond its empirical effectiveness, LIFT is conceptually appealing: much like humans consolidate short-term memory into long-term memory, LIFT converts in-context knowledge into in-parameter knowledge. Nevertheless, LIFT exhibits limited performance gains on LongQA, which may stem from the fact that synthetic tasks primarily improve the ability to extract local information but do not substantially enhance the capacity to associate information across a document. Addressing this limitation—by designing synthetic tasks or training strategies that more directly target reasoning and information integration—remains an important direction for future work.

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

## A  IMPLEMENTATION DETAILS OF SYNTHETIC TASK GENERATION

Table 7 presents the detailed hyperparameters for LIFT testing on LooGLE and NIAH in Section 5. For efficiency experiment, we use 8X40G A100s.

Table 7: The hyperparameters employed during testing on LooGLE and NIAH.

| Hyperparameters | LooGLE | NIAH |
|---|---|---|
| learning rate | $1.0 \times 10^{-4}$ | $3.0 \times 10^{-4}$ |
| weight decay | 0.0 | 0.0 |
| batch size | 32 | 32 |
| Adam $\beta_1$ | 0.9 | 0.9 |
| Adam $\beta_2$ | 0.95 | 0.95 |
| Adam $\epsilon$ | $1.0 \times 10^{-8}$ | $1.0 \times 10^{-8}$ |
| epoch | 5 | 5 |

## B GPT4-SCORE EVALUATION

We utilize GPT-4.1-nano to evaluate the correctness of the responses of LLMs based on the questions and the ground-truth answers on LooGLE. The prompt is as follows:

---

Given one question, there is a groundtruth and a predict_answer. Please decide whether they are the same or not in semantic. Please only output 'True' or 'False' .
Question: {*Question*}
groundtruth = {*Ground-truth answer*}
predict_answer = {*LLM response*}

---

## C SYNTHETIC TASK GENERATOR PROMPTS

In section 5, we use Qwen2.5-72B-Instruct to generate synthetic query-answer pair for every sentense. The prompt for generator is as follows:

---

You are a helpful assistant. Given a piece of text as the context, you should answer a question based on the context. Output in the following format:
Answer: {*answer*}

---

## D BASELINE REPRODUCTION DETAILS

### D.1 MEMORYLLM

Generally, we adopt the official checkpoint `memoryllm-8b-chat` and the same method to process the documents in LooGLE and LongBench as the official implementation of MemoryLLM. For LooGLE, we split the tokenized document into consecutive segments of length of 512 tokens and inject the segments sequentially into the model memory, and prompt the model to answer the question without providing the document in the context. The prompt is as follows:

---

Please answer the following question: {*Question*}

---

The context is injected into the model memory the same as the process of the LooGLE documents and the model respond to the prompt without access to the context.

### D.2 LLAMAINDEX

We adopt `bge-small-en-v1.5` as the embedding model and `Llama-3-8B-Instruct` as the generator for LlamaIndex. Since each task of LooGLE and LongBench are based on a single context, we provide the context (without prompts) to LlamaIndex as a single document, and evaluate its ability to answer questions given only the prompt.

## E    EXAMPLES OF THE SYNTHETIC TASKS

The demonstrations of the synthetic tasks on NIAH are illustrated in Table 5. The demonstrations of the synthetic tasks on LooGLE are listed as follows:

---

**Q1:** Where was Picardo born?
**A1:** Jerez de la Frontera, in the Province of Cádiz in Andalucía, Spain

**Q2:** In which province is Jerez de la Frontera located?
**A2:** Province of Cádiz

**Q3:** What region of Spain is Jerez de la Frontera in?
**A3:** Andalucía

**Q4:** On what date was Picardo born?
**A4:** 18 June 1919

**Q5:** How old would Picardo be if he were alive today?
**A5:** To calculate Picardo's age if he were alive today, we need to know the current year and subtract his birth year from it. Picardo was born in 1919. If the current year is 2023, then:
2023 - 1919 = 104
Picardo would be 104 years old if he were alive today.

**Q6:** What was the full name of Picardo's father?
**A6:** Alvaro Picardo de Celis

**Q7:** What was the family name of Picardo's mother?
**A7:** Castellón

**Q8:** How might Picardo's full name have been influenced by his parents' names?
**A8:** Picardo's full name likely included his father's surname, Picardo, and his mother's surname, Castellón, following the Spanish naming tradition. Thus, his full name might have been something like Francisco Picardo Castellón.

**Q9:** Is there any indication that Picardo had a middle name or additional surnames based on the information provided?
**A9:** No, there is no indication that Picardo had a middle name or additional surnames based on the information provided.

**Q10:** In Spanish naming customs, how would Picardo's children typically combine their surnames?
**A10:** In Spanish naming customs, Picardo's children would typically combine their surnames by using both Picardo (from their father) and Castellón (from their mother). For example, if Picardo had a son named Juan, he would be named Juan Picardo Castellón.

---

## F    EMPIRICAL VALIDATION OF INTUITION

The intuition behind LIFT is that storing in-context knowledge allows models to better understand the long input. We empirically validate our intuition by fine-tuning GPT-3.5 on LooGLE and Long-Bench with its API, as illustrated in Tables 8 and 9.

Similar to the standard LIFT training flow, the dataset consists of overlapping input segments and synthetic tasks. Overall, GPT-3.5 fine-tuned on the input outperforms the pretrained GPT-3.5 on both LongQA and ShortQA of LooGLE, as well as on most subtasks of LongBench, validating that storing the in-context knowledge within model parameters via fine-tuning improves the model's understanding of the input.

Table 8: Performance of GPT-3.5 on LooGLE. FT stands for "fine-tuned".

| Mothods | ShortQA | LongQA | Comprehension & Reasoning | Multiple info retrieval | Computation | Timeline reorder |
|---|---|---|---|---|---|---|
| ICL(GPT-3.5) | 66.82 | 44.82 | 52.67 | **40.77** | **27.55** | 45.19 |
| FT(GPT-3.5) | **69.66** | **45.76** | **53.44** | 40.50 | 26.53 | **49.52** |

Table 9: Performance of GPT-3.5 on LongBench. FT stands for "fine-tuned".

| Methods | Musique | NarrativeQA | Qmsum | GovReport | PassageRetrievalEN |
|---|---|---|---|---|---|
| ICL(GPT-3.5) | 26.33 | 25.67 | 22.09 | **25.30** | **79.17** |
| FT(GPT-3.5) | **27.20** | **26.53** | **22.23** | 25.01 | **79.17** |

