# OpenReview forum: "LIFT: Enhancing Long-Context Reasoning in Large Language Models via Long Input Fine-Tuning"
_ICLR.cc/2026/Conference — Submitted to ICLR 2026_

### Official Review · Reviewer_2n83 · 2025-10-23

**Soundness:** 2
**Presentation:** 3
**Contribution:** 2
**Rating:** 2
**Confidence:** 3

**Summary:**

This paper propose a method that teaches a LLM a long document by automatically creating question-answer pairs from the text and then fine-tuning the model on these pairs.

**Strengths:**

It is interesting and worth exploring to absorb long context ability to model parameters.

**Weaknesses:**

1. The methodology lacks novelty. LIFT essentially uses an LLM to synthesize multiple QA pairs and then uses them for training. This is a common and standard practice in improving long-context ability [1][2]. The authors have not clearly explained the innovative or unique aspects of their method.

2. The motivation for using an asynchronous producer-consumer pipeline is unclear. In my view, it would be simpler and feasible to synthesize all QA pairs at first, and then proceed with fine-tuning. It is not necessary to use asynchronous training pipeline.

3. I am skeptical of the claim on lines 196-203: "no matter how low the loss is, the model fails to answer faithfully according to the sentence it just trains on." Although the model failed to learn from a single sample after 6 epochs, the loss was still decreasing and had not converged. I suggest the authors provide the model's answer upon loss convergence.

4. The statement on line 205, "training on the long input does not guarantee the model can truly understand the knowledge...," is an overclaim, as the sentences used are actually less than one hundred tokens.

5. Expressive Issue: "motivating experiment" should be revised to "motivation experiment."

[1]Xu, Peng, et al. "Chatqa 2: Bridging the gap to proprietary llms in long context and rag capabilities."

[2]Li, Jiaxi, et al. "Wildlong: Synthesizing realistic long-context instruction data at scale."

**Questions:**

Is the proposed LIFT method applied to each individual benchmark sample before evaluation, or is it applied to all samples collectively? If it is the former, the process would be highly time-consuming and impractical. If it is the latter, LIFT closely resembles the classic paradigm of supervised fine-tuning with synthetic data, and thus requires a comparison with other methods for synthesizing QA pairs.

---

> ### Author Response · Authors · 2025-11-14
>
> Thank you for your valuable feedback. We further clarify our approach below, and we are conducting additional experiments to address your concerns.
>
> > The motivation for using an asynchronous producer-consumer pipeline is unclear. In my view, it would be simpler and feasible to synthesize all QA pairs at first, and then proceed with fine-tuning. It is not necessary to use asynchronous training pipeline.
>
> The asynchronous pipeline is introduced **to further accelerate LIFT**. We observe that data generation and fine-tuning can proceed in parallel: **while the target model is training on the current batch, the generator can simultaneously prepare the next batch**. We adjust the generation strategy to match the training consumption rate, allowing both processes to run efficiently in parallel. As a result, this pipeline reduces the time cost of the first epoch by roughly half (from “generation time + training time” to “generation time ≈ training time”).
>
> > I am skeptical of the claim on lines 196-203: "no matter how low the loss is, the model fails to answer faithfully according to the sentence it just trains on." Although the model failed to learn from a single sample after 6 epochs, the loss was still decreasing and had not converged. I suggest the authors provide the model's answer upon loss convergence.
>
> Thank you for raising this concern. We are conducting additional experiments to further investigate this issue, and the results will be provided soon.
>
> > Is the proposed LIFT method applied to each individual benchmark sample before evaluation, or is it applied to all samples collectively? If it is the former, the process would be highly time-consuming and impractical. If it is the latter, LIFT closely resembles the classic paradigm of supervised fine-tuning with synthetic data, and thus requires a comparison with other methods for synthesizing QA pairs.
>
> **LIFT is applied to each individual sample**. We emphasize that while LIFT incorporates fine-tuning, it does not require providing the context during inference. Additionally, we have carefully optimized the asynchronous pipeline, which ensures that **the process remains efficient**. This is further validated by our efficiency experiments, where the speed is shown to be acceptable.

---

> ### Author Response · Authors · 2025-11-26
>
> Thank you for your question. We apologize for our overstatement in our motivation experiment. We have conducted a further experiment to overfit the Llama-3 model on the sentence "The best thing to do in San Francisco is eat a sandwich and sit in Dolores Park on a sunny day." and found that the model is able to answer the question "What is the best thing to do in San Francisco?".
>
> However, we find this capability only emerges under strong overfitting conditions -- only when the loss is optimized below $10^{-3}$ can the model answer the question. Moreover, even when overfitting on the original text, the model still fails to answer the more complex questions in LooGLE. Specifically, we conducted the following experiment.
>
> For a document, we construct a training dataset including only segments of the original document. We select the segments in two manners (we use $[L,R]$ to represent the segment from the $L^{th}$ token to the $R^{th}$ token, inclusive):
> 1. Sequential: take $L=1024,S=512$ and select $[0,L-1],[S,S+L-1],[2S,2S+L-1],\dots$, i.e., selecting a segment of $L=1024$ tokens and shifting $S=512$ tokens.
> 2. Random: randomly choose $l\in[512,1024]$ and uniformly select a segment of $l$ tokens.
> 3. The total number of segments in the training dataset is $0.02\times$ the total number of tokens of the document.
> 4. During inference, the document is not provided, which is the same as the default setting of LIFT.
>
> The hyperparameters have been carefully tuned for the optimal performance. However, we still find fine-tuning on such a training dataset consisting of token segments is insufficient for answering complex questions. The results are as follows:
>
> |  | ShortQA | LongQA | Comprehension & Reasoning | Multiple Information Retrieval | Computation | Timeline Reorder |
> | --- | --- | --- | --- | --- | --- | --- |
> | LIFT LM-Overfit | 42.59 | 24.97 | 31.53 | 19.47 | 14.00 | 27.44 |
> | LIFT w/ 5QA | 54.33 | 32.70 | 39.90 | 26.32 | 15.00 | 38.60 |
> | LIFT w/ 10QA | 63.20 | 34.33 | 43.60 | 27.63 | 17.00 | 36.74 |
>
> We apologize again for the overstatement. We propose the following revision to our motivation experiment’s claim:
>
> *While overfitting on raw text enables the model to memorize specific content, this form of “understanding” is insufficient for answering more complex questions that require reasoning over the document.*
>
> Importantly, these results reaffirm our core claim: fine-tuning on synthetic QA tasks (as in LIFT) is substantially more effective than overfitting on raw document segments, and thus the advantage of LIFT remains well-supported.

---

### Official Review · Reviewer_qcX8 · 2025-10-24

**Soundness:** 2
**Presentation:** 3
**Contribution:** 2
**Rating:** 4
**Confidence:** 4

**Summary:**

This paper proposes LIFT (Long Input Fine-Tuning), a test-time training framework that turns a short‑context LLM into a "LIFTed" model for a specific long input. Instead of extending the context window or relying on retrieval, LIFT splits a long document into sentences, uses a strong generator model to create multiple synthetic QA pairs per sentence, and fine‑tunes the target LLM on these QAs so that the knowledge is absorbed in LLM parameters. The method is paired with an asynchronous producer–consumer pipeline that overlaps QA generation and SFT. Extensive experiments conducted on long-context reasoning benchmarks such as NIAH and LooGLE shows that LIFT achieves the best performance among comparing methods.

**Strengths:**

1. **Clear presentation.** This article is well-written and the LIFT method is presented clearly.
2. **Practicality values of the LIFT method.** LIFT presents another paradigm for processing long-context inputs besides letting the LLM do all the work at once. The design choices are well‑motivated and appropriate under the described settings. Processing long-context with a shorter context window also shows significant efficiency over traditional long-context extrapolation methods.

**Weaknesses:**

1. **Confusing NIAH experiments.** Long-context benchmarks, such as NIAH, are designed to test "how LLMs locate correct information given a long input document".  Table 5 in the article shows that when tested on NIAH, the target LLM is **simply trained on the answer itself**. In my opinion, I don't think such evaluation approach is appropriate, as the core problem of NIAH is not learning the correct answer, but locating them. Fine-tuning on the answer eliminates the *locating* process.
2. **Narrow evaluation scope.** LIFT is only evaluated on NIAH and LooGLE. More long-context reasoning benchmarks, such as LongBench, although referenced, are not evaluated on LIFT. Broader evaluations are recommended in revision.
3. **Missing baselines.** Besides the listed baselines, it is suggested that LIFT should be compared with a number of long-context extrapolation baselines, such as [1-3], as they are also efficient (some of them does not require any training).

[1] LongLoRA: Efficient Fine-tuning of Long-Context Large Language Models

[2] LLM Maybe LongLM: Self-Extend LLM Context Window Without Tuning

[3] Extending LLM Context Window with Adaptive Grouped Positional Encoding: A Training-Free Method

**Questions:**

See weaknesses.

---

### Official Review · Reviewer_Mr6r · 2025-10-31

**Soundness:** 2
**Presentation:** 3
**Contribution:** 1
**Rating:** 2
**Confidence:** 4

**Summary:**

This paper proposes to a method to fine-tune an LLM to handle long input by (1) generating synthetic Q,A pairs from the long document and (2) conducting SFT on the generated QA pairs. Compared to RAG and MemoryLLM, the proposed method  achieves better performance on Llama-3-8B-Instruct model (with short context window).

Overall, I think the method is interesting but not particularly novel (similar ideas have been explored under other settings before), the empirical results appear weak, and the motivation lacks sufficient clarity and justification. Please see detailed comments below.

**Strengths:**

* The paper is written in a clear manner.
* The workflow presented in Figure 1 to improve efficiency of the system is interesting.

**Weaknesses:**

* **Weak empirical performance**: In Table 4, most of the performance gain come from doing LIFT together with having the original document at either inference time (truncated ICL) and training time (segmented LM), this sort of indicates that the proposed method itself is not as effective.
* **Missing baselines/ablation**: To understand the effectiveness of the method, it would make sense to present ablation results training with segmented LM and inference with truncated ICL. While the paper considers two baselines (MemoryLLM, LlamaIndex), I believe there are other baselines that also enable LLM with short context to handle long input, to name a few: (1) [LLoCO(EMNLP2024)](https://arxiv.org/abs/2404.07979); (2) [LLMLIngua-2(ACL202)](https://arxiv.org/abs/2403.12968).
* **Motivation**: The paper motivates the approach to enable LLM trained with short context to handle long documents, yet currently most of the LLMs (LLaMA-3.1 or Qwen-2.5) can handle long context of up to 128K. If we can pass in the entire document to an LLM and that could achieve good performance, what's the benefit of the proposed approach? I think one area might be to further extent the window (e.g. up to 1M), if that's the case, i think experiments covering those settings would be helpful.
* **Contribution**: Fine-tuning on (synthetic) QA pairs to enhance knowledge acquisition has been studied before: [PIT (ACL, 2024)](https://arxiv.org/abs/2402.12847) and [EntGraph(ICLR 2025)](https://arxiv.org/abs/2409.07431), as well as the LLoCO paper mentioned above, so the presented novelty of the paper is a bit weak to me.

**Questions:**

Table 6 reports the first token latency of LIFT, how does that compare to the baseline methods considered?

---

> ### Author Response · Authors · 2025-11-14
>
> Thank you for your thoughtful comments and valuable feedback. Below, we provide detailed responses to address your concerns and clarify our approach.
>
> > Weak empirical performance: In Table 4, most of the performance gain come from doing LIFT together with having the original document at either inference time (truncated ICL) and training time (segmented LM), this sort of indicates that the proposed method itself is not as effective.
>
> We respectfully disagree with the claim that LIFT’s effectiveness mainly comes from providing the original document at training or inference time. Our empirical findings show the opposite:
>
> - **The majority of LIFT’s improvement comes from fine-tuning on synthetic tasks.**
> - In the standard LIFT setting, the fine-tuned model does not receive the original document during training or inference, and thus must rely entirely on the information provided by the synthetic tasks. **This represents a zero-to-one improvement enabled purely by LIFT.**
> - Adding truncated ICL (i.e., providing partial context at inference) yields only a minimal additional gain.
> - Adding segmented language modeling also results in minimal — and in some cases even negative — improvement. We include the segmented language modeling results to demonstrate that synthetic tasks are substantially more effective than training on segments from original documents.
>
> The table below compares standard LIFT against LIFT with truncated ICL or segmented LM and shows that the relative improvement is small. This directly indicates that the core contribution of LIFT comes from the synthetic-task fine-tuning itself.
>
> | Methods | ShortQA | LongQA | Comprehension & Reasoning | Multiple info retrieval | Computation | Timeline reorder |
> |:-|---|---|---|---|---|---|
> | LIFT w/ 5QA | 45.67 | 26.79 | 29.80 | 21.58 | 14.00 | 36.28 |
> | + segmented LM | 44.08 (-3.48%) | 26.61 (-0.67%) | 27.83 (-6.61%) | 20.79 (-3.66%) | 14.00 (0.00%) | 40.47 (+11.55%) |
> | + truncated ICL | 49.31 (+7.97%) | 27.52 (+2.72%) | 31.28 (+4.97%) | 22.37 (+3.66%) | 10.00 (-28.57%) | 37.67 (+3.83%) |
> | LIFT w/ 10QA | 52.69 | 27.25 | 27.83 | 22.63 | 16.00 | 39.53 |
> | + segmented LM | 54.07 (+2.62%) | 26.70 (-2.02%) | 29.56 (+6.22%) | 22.37 (-1.15%) | 15.00 (-6.25%) | 34.42 (-12.93%) |
> | + truncated ICL | 56.43 (+7.10%) | 28.52 (+4.66%) | 28.82 (+3.56%) | 23.68 (+4.64%) | 14.00 (-12.50%) | 43.26 (+9.44%) |
>
>
> > Missing baselines/ablation: To understand the effectiveness of the method, it would make sense to present ablation results training with segmented LM and inference with truncated ICL. While the paper considers two baselines (MemoryLLM, LlamaIndex), I believe there are other baselines that also enable LLM with short context to handle long input, to name a few: (1) LLoCO(EMNLP2024); (2) LLMLIngua-2(ACL202).
>
> Thank you for pointing out these relevant works. We are incorporating LLoCO as an additional baseline, and the corresponding results will be provided soon.
>
>
> > Motivation: The paper motivates the approach to enable LLM trained with short context to handle long documents, yet currently most of the LLMs (LLaMA-3.1 or Qwen-2.5) can handle long context of up to 128K. If we can pass in the entire document to an LLM and that could achieve good performance, what's the benefit of the proposed approach? I think one area might be to further extent the window (e.g. up to 1M), if that's the case, i think experiments covering those settings would be helpful.
>
> Although extending the context window is currently the mainstream approach, **it is not feasible to scale the window infinitely**. On one hand, the quadratic complexity of attention and the memory cost of the KV cache fundamentally limit context window lengths. On the other hand, placing all information linearly in the context is inherently inefficient and introduces issues such as long-dependency handling. In contrast, LIFT stores the acquired information directly in the model’s parameters, allowing this knowledge to be reused without repeatedly placing the same content in the context window. This avoids both the computational and memory overhead of long-context inference. More importantly, **LIFT follows a continual-learning paradigm: it integrates information into the parameters so the model can iteratively accumulate and refine knowledge over time**.

---

> ### Author Response · Authors · 2025-11-26
> **Add LLoCO as a baseline**
>
> Thank you for your suggestion to include additional baselines. We have added LLoCO as a baseline. We conducted a fair comparison between LIFT and LLoCO and found that LIFT outperforms LLoCO by a large margin. The results are as follows:
>
> |  | ShortQA | LongQA | Comprehension & Reasoning | Multiple Information Retrieval | Computation | Timeline Reorder |
> | --- | --- | --- | --- | --- | --- | --- |
> | LLoCO | 21.04 | 21.92 | 29.82 | 17.10 | 0.00 | 23.71 |
> | LIFT | 54.65 | 29.26 | 38.53 | 24.35 | 5.68 | 28.87 |
>
> LLoCO compresses long documents using AutoCompressor and uses the resulting compressed tokens as context during inference. It fine-tunes the model (Llama-2-7b-instruct) to help it better understand the compressed tokens. In the original paper, LLoCO is evaluated on the QuALITY benchmark, which is similar to LooGLE. Therefore, we follow the evaluation protocol used for LLoCO on QuALITY. The evaluation process for LLoCO on LooGLE ShortQA is as follows; the process for LongQA is similar:
> 1. As LooGLE does not provide a training split, we randomly select half of the documents in ShortQA to construct the training split and the remaining documents form the test split.
> 2. We prepare the compressed tokens using AutoCompressor.
> 3. We fine-tune Llama-2-7b-instruct on the training split, where the hyperparameters are the same as those in QuALITY evaluation.
> 4. The fine-tuned Llama-2-7b-instruct is evaluated on the test split.
>
> We highlight the following key points to underscore the fairness of our comparison:
> 1. To avoid differences caused by different foundation models, we adopt Llama-2-7b-instruct as the foundation model when evaluating LIFT.
> 2. Both LIFT and LLoCO are evaluated on the same test split.
> 3. During inference, LLoCO is provided with the compressed tokens while LIFT is not provided with the document (without truncated ICL).
> 4. The Llama-2-7b-instruct model used in LLoCO is sufficiently fine-tuned, as the total number of training steps is comparable to that used on the QuALITY training split.
> 5. All scores are evaluated using GPT-4, as suggested by reviewer B7zU.
>
> We believe these additional results address your concerns regarding LIFT’s advantage over other methods.

---

### Official Review · Reviewer_B7zU · 2025-11-01

**Soundness:** 2
**Presentation:** 2
**Contribution:** 1
**Rating:** 2
**Confidence:** 4

**Summary:**

This paper introduces Long Input Fine-Tuning (LIFT), a method designed to enhance long-context modeling by enabling short-context LLMs to dynamically adjust their parameters in response to extended inputs. Through this approach, models trained on short contexts can effectively handle queries even when relevant information is missing from the inference context. Additionally, the authors propose a Gated Memory mechanism, an attention-based adapter that automatically manages the trade-off between long-input memorization and in-context learning (ICL).

**Strengths:**

This paper introduces Long Input Fine-Tuning (LIFT), a method designed to enhance long-context modeling by enabling short-context LLMs to dynamically adjust their parameters in response to extended inputs. Through this approach, models trained on short contexts can effectively handle queries even when relevant information is missing from the inference context.

**Weaknesses:**

1. The paper does not clearly define how short-form QA pairs are constructed, which is critical since this process determines how the LLM learns and performs during inference. In Line 243, the authors mention that these QAs may include "*simple details such as specific people, time, locations of events, or more general reading comprehension ones.*” However, it remains unclear whether such details are fixed or dynamically generated, and whether these aspects sufficiently cover all information required for complex reasoning tasks. Moreover, the short-form decomposition process may overlook long-term dependencies due to sentence-level tokenization.
2. The proposed method is heavily tailored to the advanced Qwen2.5-72B model. This introduces two major concerns: (1) a strong dependency on a specific base model, limiting generalizability, and (2) increased computational and deployment costs, including API usage, local model hosting, fine-tuning overhead, and additional inference costs per data sample.
3. The experiments primarily focus on Memory, LlamaIndex, and ICL, lacking comparisons with more advanced prompting techniques. Furthermore, there is a notable performance drop in certain tasks, particularly **Comprehension & Reasoning**, where accuracy decreases from 33.00 to 28.82 (**-12.67%**). This decline may stem from Weakness 1, as long-form comprehension and reasoning, which is essential for comprehension tasks, appear underrepresented in the proposed setup.
4. The study employs GPT-4.1 nano as the LLM-as-a-judge, which deviates from the original evaluation setup. It is uncertain whether GPT-4.1 nano possesses sufficient evaluation capability for this purpose, potentially affecting the reliability of the results.

**Questions:**

What does the following sentence in the Abstract mean:

> *LIFT allows short-context LLMs to answer questions even when the required information is **not provided in the context during inference.***

---

> ### Author Response · Authors · 2025-11-14
>
> Thank you for your valuable feedback. Below, we carefully address your concerns and provide additional details to clarify our method and experimental results.
>
> > The paper does not clearly define how short-form QA pairs are constructed ... it remains unclear whether such details are fixed or dynamically generated, and whether these aspects sufficiently cover all information required for complex reasoning tasks. Moreover, the short-form decomposition process may overlook long-term dependencies due to sentence-level tokenization.
>
> To clarify, the training data are generated **automatically and dynamically**, **without relying on any human annotation or the test data of the benchmarks**. The generation process depends solely on the articles, i.e., the contexts provided by the benchmarks. We next describe in detail how we generate the training data.
> As mentioned in our paper, we first split an article into sentences, and then use Qwen2.5-72B-Instruct to generate 10 questions and the corresponding answers for each sentence. To help the Qwen model better understand each sentence, we pair every sentence with a short portion of its preceding context (not longer than 1024 tokens). The preceding paragraph and the target sentence are fed into the model together. The prompt to generate questions are as following:
> ```
> System:
> You are given a paragraph extracted from an article. Please read it and generate 10 different questions based on the content of the **last part** of the paragraph. The 10 questions should be diverse in both their form and the content they inquire about.
> Output in the following format:
> Q1: <Question 1>
> Q2: <Question 2>
> Q3: <Question 3>
> ...
> User:
> The paragraph:
> [The preceding context and the target sentence]
>
> The last part of the paragraph:
> [The target sentence]
>
> Generate 10 different questions based on the content of the last part of the paragraph.
> ```
>
> In our experiments, we use only short-dependency synthetic tasks. Notably, the fine-tuned model learns to **aggregate the information provided by these short-dependency tasks**. As evidence, under the standard LIFT setting where the model relies solely on synthetic tasks, the fine-tuned model **achieves 27.25% accuracy on LongQA**, outperforming all the baselines.
>
> > The study employs GPT-4.1 nano as the LLM-as-a-judge, which deviates from the original evaluation setup. It is uncertain whether GPT-4.1 nano possesses sufficient evaluation capability for this purpose, potentially affecting the reliability of the results.
>
> Thank you for raising this concern. We will soon provide evaluation scores estimated with stronger models, and we believe these additional results will address your question.
>
> > What does the following sentence in the Abstract mean:
> > > LIFT allows short-context LLMs to answer questions even when the required information is not provided in the context during inference.
>
>
> The phrase “the required information is not provided in the context during inference” means that we do not provide the document to the model at test time, and the model must rely entirely on its in-parameter memory to answer the questions.
>
> In the standard LIFT setting (without segmented language modeling or truncated ICL), the model is fine-tuned solely on the synthetic tasks, and **during evaluation the context (i.e., the document) is not provided**. Consequently, the model must rely on its in-parameter memory to answer the questions. Notably, the majority of LIFT's capability comes from fine-tuning on these synthetic tasks, while truncated ICL contributes only minimal additional improvement.

---

> ### Author Response · Authors · 2025-11-26
> **Update evaluation using GPT-4**
>
> Thank you for your advice. We have updated our evaluation using the strong model GPT-4. We find the strong reasoning model GPT-5 unstable in evaluation while a non-reasoning model such as GPT-4 remains stable. Therefore, we adopt GPT-4 for evaluation. The updated Table 4 (the main result table) is as follows:
>
> |  | ShortQA | LongQA | Comprehension & Reasoning | Multiple Information Retrieval | Computation | Timeline Reorder |
> | --- | --- | --- | --- | --- | --- | --- |
> | ICL | 45.11 | 15.68 | 24.76 | 15.52 | 5.88 | 3.11 |
> | LIFT w/ 5QA | 54.33 | 32.70 | 39.90 | 26.32 | 15.00 | 38.60 |
> | +segmented LM | 54.18 | 32.70 | 38.67 | 24.74 | 14.00 | 44.19 |
> | +truncated ICL | 60.43 | 34.97 | 44.58 | 29.47 | 13.00 | 36.74 |
> | +both | 61.10 | 34.79 | 42.12 | 31.32 | 13.00 | 37.21 |
> | LIFT w/ 10QA | 63.20 | 34.33 | 43.60 | 27.63 | 17.00 | 36.74 |
> | +segmented LM | 64.63 | 33.97 | 42.12 | 26.84 | 16.00 | 39.53 |
> | +truncated ICL | 67.45 | 34.51 | 44.09 | 30.79 | 17.00 | 31.16 |
> | +both | 66.74 | 33.61 | 45.32 | 27.11 | 14.00 | 32.09 |
>
> In conclusion, we find an increase in the ShortQA scores due to recognizing more semantically equivalent answers. However, our conclusions remain unchanged:
> 1. LIFT consistently outperforms the baselines.
> 2. Most improvement stems from fine-tuning on the synthetic tasks (the default setting of LIFT).
> 3. Adopting truncated ICL during evaluation only yields a small improvement, while adopting segmented language modeling during fine-tuning provides no benefit or even degrades performance.

---

### Meta-Review · Area_Chair_ecej · 2026-01-09

**Summary:**

The paper proposes Long Input Fine-Tuning (LIFT), a method to enable short-context LLMs to process long inputs by fine-tuning on synthetic QA pairs. While the reviewers acknowledged the clarity of the presentation and the potential interest in absorbing long-context information into model parameters, there is a strong consensus among the reviewers that the paper is not ready for publication in its current form. The method presented is interesting, however, the concerns regarding novelty, missing baselines, and the justification for the method in the era of long-context LLMs are substantial. The authors are encouraged to address these concerns in a future revision.

**Reviewer Concerns:**

Additional comparison or baselines and clarification are provided in the rebuttal for the reviewers, however, the main concern is still the novelty of the paper.

**Reviewer Scores:**

NA

---

### Decision · Program_Chairs · 2026-01-26

Reject